# Comprehensive Contact Tracing, Testing and Sequencing Show Limited Transmission of SARS-CoV-2 between Children in Schools in Norway, August 2020 to May 2021

**DOI:** 10.3390/microorganisms9122587

**Published:** 2021-12-14

**Authors:** Brita Askeland Winje, Trine Skogset Ofitserova, Ola Brønstad Brynildsrud, Margrethe Greve-Isdahl, Karoline Bragstad, Rikard Rykkvin, Olav Hungnes, Hilde Marie Lund, Karin Nygård, Hinta Meijerink, Lin Thorstensen Brandal

**Affiliations:** 1Department of Infection Control and Vaccine, Norwegian Institute of Public Health, 0213 Oslo, Norway; TrineSkogset.Ofitserova@fhi.no (T.S.O.); Margrethe.Greve-Isdahl@fhi.no (M.G.-I.); Hinta.Meijerink@fhi.no (H.M.); 2Faculty of Health Sciences, Oslo Metropolitan University, 0167 Oslo, Norway; 3Department of Method Development and Analytics, Norwegian Institute of Public Health, 0213 Oslo, Norway; ola.brynildsrud@fhi.no; 4Department of Virology, Norwegian Institute of Public Health, 0213 Oslo, Norway; Karoline.Bragstad@fhi.no (K.B.); Rikard.Rykkvin@fhi.no (R.R.); Olav.Hungnes@fhi.no (O.H.); 5Department of Infection Control and Preparedness, Norwegian Institute of Public Health, 0213 Oslo, Norway; HildeMarie.Lund@fhi.no (H.M.L.); KarinMaria.Nygard@fhi.no (K.N.); LinCathrineT.Brandal@fhi.no (L.T.B.); 6European Program for Public Health Microbiology Training (EUPHEM), European Centre for Disease Prevention and Control, (ECDC), 169 73 Solna, Sweden

**Keywords:** SARS-CoV-2, COVID-19, child, contact, transmission, whole genome sequencing, infection prevention and control

## Abstract

The role of children in the spread of severe acute respiratory syndrome coronavirus 2 (SARS-CoV-2) in schools has been a topic of controversy. In this study among school contacts of SARS-CoV-2 positive children in 43 contact-investigations, we investigated SARS-CoV-2 transmission in Norway, August 2020–May 2021. All participants were tested twice within seven to ten days, using SARS-CoV-2 PCR on home-sampled saliva. Positive samples were whole genome sequenced. Among the 559 child contacts, eight tested positive (1.4%, 95% CI 0.62–2.80), with no significant difference between primary (1.0%, 95% CI 0.27–2.53) and secondary schools (2.6%, 95% CI 0.70–6.39), *p* = 0.229, nor by viral strain, non-Alpha (1.4%, 95% CI 0.50–2.94) and Alpha variant (B.1.1.7) (1.7%, 95% CI 0.21–5.99), *p* = 0.665. One adult contact (1/100) tested positive. In 34 index cases, we detected 13 different SARS-CoV-2 Pango lineage variants, with B.1.1.7 being most frequent. In the eight contact-investigations with SARS-CoV-2 positive contacts, four had the same sequence identity as the index, one had no relation, and three were inconclusive. With mitigation measures in place, the spread of SARS-CoV-2 from children in schools is limited. By excluding contact-investigations with adult cases known at the time of enrolment, our data provide a valid estimate on the role of children in the transmission of SARS-CoV-2 in schools.

## 1. Introduction

Many countries adopted school closures and strict infection prevention and control measures (IPC) to limit the spread of SARS-CoV-2, based on the assumption that transmission among children may augment the spread of the virus in the wider community. The high societal costs related to these interventions have invited questions as to whether they are justified, given that children are mostly mildly affected by the virus and mortality rates are very low [1,2].

Children of all ages are susceptible to and may transmit SARS-CoV-2 [3,4], although younger children seem to transmit SARS-CoV-2 less frequently than older children in school settings [3,5]. At the outset of this study, little was known about the role of children in SARS-CoV-2 transmission in schools, specifically the role of asymptomatic infections [6]. Since then, several contact-investigation studies have been published, including a subset of the data included in this report [7], showing limited transmission in schools when appropriate mitigation measures are in place [5]. Yet, the role of children in school transmission is still not completely understood. Asymptomatic and mild symptoms in children may have led to low case-ascertainment of infections in children, specifically early in the pandemic [8]. In addition, the interplay between the school and community incidence challenges the interpretation of data [5], and super-spreading events have also been reported [9].

In Norway, children’s daycare and schools were closed on 13 March 2020, and reopened under strict IPC measures from 20 April 2020. A traffic light model was developed to guide school administrators on IPC strategies [10]. This three-tiered system, with non-pharmaceutical measures, depends on local incidence and infection pressure. The guidelines advised the establishment of cohorts consisting of small permanent groups of children and staff with limited interaction between cohorts, alongside timely testing and isolation of symptomatic cases, and tracing and quarantine of their contacts. Until January 2021, the recommended quarantine period was ten days since the day of last contact, thereafter seven days provided a negative test. The red level indicated smaller cohorts and more restrictive measures [10].

While high vaccination coverage in adults will limit transmission and disease burden, vaccines for younger children are not yet approved. Understanding children’s role in transmission is therefore still important for future mitigation strategies. Feasibility and high sensitivity and specificity of saliva for detection of SARS-CoV-2 by PCR makes this an attractive means for testing of children [11]. To improve knowledge on child-to-child and child-to-adult SARS-CoV-2 transmission in school-settings, we used home-sampled saliva to investigate how many contacts were infected with SARS-CoV-2 after contact with a child with a confirmed SARS-CoV-2 infection at school and whether there was a difference in transmission by age and viral variants.

## 2. Materials and Methods

Between August 2020 and the end of May 2021, we prospectively followed school contacts of 43 children with laboratory-confirmed SARS-CoV-2 (index cases) who had physically been attending school within 48 h prior to symptom onset or the sampling date for asymptomatic cases. The study included primary (1–7th grade) and secondary (8–10th grade) schools in Oslo and Viken counties. We obtained two self-collected saliva samples from all contacts, at least four days apart, within the seven- to ten-day-long mandatory quarantine period. We excluded contact investigations around child cases in schools with known SARS-CoV-2 infected adults at the time of enrolment. The study was conducted in parallel to measures taken by the local health services.

Local health authorities identified school contacts as per national guidelines. We used the schools’ digital platform to send study invitations to the same school contacts and the index, with one same-day reminder. The school administrations were not involved in any other study procedures. Contacts outside of school were not included. Consent was obtained from all participants aged 16 years or above, or from both parents for participants aged <16 years. In case of sole parental responsibility, a single consent was sufficient.

### 2.1. Definitions

The index was the child first identified with confirmed SARS-CoV-2 leading to the contact-investigation. A primary case was a contact of the index, who tested positive for SARS-CoV-2 in the first saliva sample. A secondary case was a contact who tested positive for SARS-CoV-2 in the second saliva sample, following a first negative test. A SARS-CoV-2 infected contact was a contact who tested positive in either of the two tests, thus including both the primary and secondary cases. We divided the study-period into before and after the more transmissible Alpha variant (B.1.1.7) was established in the study area (pre-Alpha; August 2020 to February 2021, Alpha; March 2021 to May 2021) [12].

### 2.2. Demographic and Exposure Data

Participants filled a baseline questionnaire including demographic, exposure and clinical information and a form for daily reporting on the presence or absence of symptoms. Non-cases also filled a questionnaire on recent exposure at the study end (available at https://www.fhi.no/en/studies/corona-child-study/, accessed on 1 December 2021).

### 2.3. Laboratory Data

At enrolment, all participants were provided with equipment and offered face-to-face and printed instruction on how to home-sample saliva for RdRp gene PCR [13]. Participants were asked to collect 1 mL saliva (not sputum) in the morning, before eating, drinking, or brushing teeth [14]; the first sample was to be collected on the morning after enrolment and the second sample on the morning after the last day of quarantine. Participants added viral transport media to the saliva to preserve the virus during transportation. A study-coordinator was available on site and offered residential delivery and pick-up services to ensure timely transportation for laboratory procedures at the National Reference laboratory of influenza and coronaviruses at the Norwegian Institute of Public Health (NIPH). All saliva samples were tested for the presence of SARS-CoV-2 by rRT-PCR at the Norwegian Institute of Public Health (NIPH). RNA was extracted from samples (200 µL) using MagNaPure 96 DNA and Viral NA Small Volume kits (No. 6543588001, Roche, Basel, Switzerland), and eluted in 50 µL. Saliva samples with too much mucus were mixed 1:1 with a sputum lysis buffer containing N-acetylcystein (10 g/L) and were shaken for 30 min. A semi-quantitative real-time reverse transcription polymerase chain reaction (rRT-PCR) was performed using the AgPath-ID One-step RT-PCR kit (No. 4387391, Life Technologies, Carlsbad, CA, USA) with primers and probes targeting two SARS-CoV-2 RdRp gene targets, developed at Institut Pasteur (Paris, France), and shared in the WHO protocol inventory [13]. A 25 µL reaction was set up, containing 5 µL of RNA. Criteria for a positive reaction were a cycle of threshold (Ct) value of less than 40 for both PCR targets and a credible amplification curve. Inconclusive results were resolved by repeating tests.

The project coordinator notified all participants about the test-results. The median number of days between the date of the last exposure (contact with the index case) and the date of the first saliva sample was four (range: three to six days). The median number of days between the two saliva samples was six (range: three to ten days).

Samples from all positive cases were whole genome sequenced using two multiplexed amplicon approaches; ARTIC-network nCoV-19 protocol v.3 [15,16] (available at https://arctic.network/ncov-2019, accessed on 1 December 2021), run on either the Illumina MiSeq platform (Illumina Inc, San Diego, CA, USA), the NanoPore GridIon (Oxford Nanopore Technologies, Oxford, UK); or the Swift Amplicon SARS-CoV-2 Panel (Swift Biosciences, Ann Arbor, MI, USA) on Illumina NovaSeq (Illumina Inc, San Diego, CA, USA) at the Norwegian Sequencing Centre, Oslo University Hospital, according to the manufacturer’s instructions with minor modifications. We used the PANGO lineages nomenclature (https://github.com/cov-lineages/pangolin, accessed on 28 September 2021) to define the SARS-CoV-2 variant [17]. Sequences were aligned in a codon-aware manner to the reference genome Wuhan-Hu-1 (GenBank accession no: MN908947.3) using the program Nextalign v.0.2.0 (available at https://github.com/nextstrain/nextclade, accessed on 1 December 2021), and a maximum-likelihood phylogenetic tree was created using IQTREE v.2.0.3 (available at https://iqtree.org, accessed on 1 December 2021) [18]. Pairwise single nucleotide polymorphism (SNP) analysis determined possible transmission in contact tracings with primary or secondary cases identified. We considered contacts with a SNP distance of 0–2 SNPs as verified transmissions [19].

For quality assessment purposes we linked the study data with the laboratory results from the national laboratory database in the Norwegian Surveillance System for Communicable Diseases (MSIS) through the participants’ unique national identity numbers. We obtained data from MSIS from the two weeks prior to the date of the first saliva sample and four weeks after the date of the second saliva sample, or the first sample when we did not receive a second sample.

### 2.4. Epidemiological Data

We obtained data on the local COVID-19 incidence rate over the last 14 days prior to date of sample collection for diagnosis of the index case in each contact-investigation. To assess possible underestimation of transmission due to non-participation we obtained aggregated data from the municipal health officer on the number of contacts with confirmed SARS-CoV-2 identified in the routinely performed contact investigation. In the municipalities, testing of all contacts was only routine in the second half of the academic year.

An overview of data and sources is presented in Appendix A
Table A1.

### 2.5. Mitigation Measures in Schools

Semi-structured phone interviews with school staff about mitigation measures were conducted in parallel with the contact investigations. Two researchers independently categorized the responses into eight broad categories (Box 1) and scored the schools’ IPC measures on a scale from 0 to 8, with higher scores indicating higher consistency with national guidelines relative to the traffic light model. The scoring system was based on expert opinions. In case of discordant opinions, a third author was included to reach consensus. Each school’s IPC-score is available in Appendix A
Table A2.

Box 1Categories of infection preventionand control measures in schools.Stay at home policy for all if symptomatic Enforced classroom hygiene Defined child cohorts, limited mixing of cohorts Limit shared spaces for children indoors Limit shared spaces for children outdoors Reduce the number of staff involved in each cohort Reduce physical contact between staff Work-from-home policy for staff whenever possible

### 2.6. Analyses

We calculated the percentages of the primary, secondary and all SARS-CoV-2 positive contacts among all contacts with 95% confidence intervals, excluding index cases both in the numerator and the denominator. For secondary cases we also excluded primary cases and contacts who did not provide a second saliva sample (n18, 3%) from the denominator. Comparison between the groups was made using Fisher’s exact test (*p*-value ≤ 0.05). Descriptive measures other than percentages are presented as median (range).

All data was analyzed using STATA, (StataCorp. LLC, Release 16, College Station, TX, USA).

## 3. Results

The study includes 43 contact investigations in Oslo and Viken counties from August 2020 until May 2021, with 559 child and 100 adult contacts, as seen in Table 1. Eleven (26%) contact investigations were included in the period in which Alpha was dominant. Overall, 60% (559/935) of child contacts and 63% (100/158) of adult contacts consented to participate in the study. The median participation for child and adult contacts was lower in the Alpha period compared to the pre-Alpha period (Table 1).

Among index cases, 79% (34) consented to participate. At the time of enrolment, 44% (15) reported to have symptoms; six reported fever with at least one other symptom and nine reported other symptoms without fever. Two index cases did not respond to the question. Among those without symptoms at enrolment, two additional indexes reported fever during follow-up.

The local incidence rate in the municipality of the Oslo borough over the two weeks prior to the initiation of the contact investigation showed large variation over time and place and ranged from 17–763 cases per 100,000. The corresponding incidence rates for Oslo and Viken counties ranged from 29–471 cases per 100,000, (Appendix A
Table A2). The IPC-measures in schools were on red level according to the traffic-light model during 40% (17) of the contact investigations, and on yellow level in the remaining cases.

### 3.1. Infection Rates

The primary and secondary infection rates remained <1% in both the pre-Alpha and the Alpha period, with overlapping confidence intervals, and at 1.4% overall (see Table 2). We found no statistically significant difference in the overall infection rates in the pre-Alpha and Alpha period (Fisher’s exact test, *p* = 0.665). Only one primary case and no secondary cases were identified in the lowest grades (1–4th). The percentage infected contacts was higher in secondary than in primary schools, but the difference was not statistically significant (*p* = 0.229), and the percentage remained low (see Table 2).

Only one adult contact (1/100, 1%) tested positive in the study with SARS-CoV-2 confirmed in the first saliva sample (primary case) in the pre- Alpha period (5th–7th grade).

### 3.2. Characteristics of the Virus

Among the 34 index cases who consented to participate, whole genome sequencing (WGS) identified 13 different SARS-CoV-2 Pango lineage variants [17] in 33 index cases (see Table 3). For one index case, WGS failed due to low viral RNA content in the sample (PCR cycle threshold Ct > 33). The B.1.1.7 (Alpha) was the most frequent variant.

Four of the nine index cases that did not consent to participate were enrolled during the Alpha-period. We assume that these contacts were exposed to the more transmissible Alpha variant, since this was the dominant variant in this period, growing from ~80% in the beginning of March to ~95% by the end of May 2021 [12].

### 3.3. Possible Transmission

Primary or secondary cases were identified in eight separate contact investigations (see Table 4). In four of the contact-investigations (ID 08; 14; 17; 24), matching SARS-CoV-2 Pango lineages were identified in index cases and the corresponding primary or secondary cases. The phylogenetic analysis showed ≤2 SNPs difference between cases in these contact investigations, consistent with direct transmission among cases (Figure 1). Transmission can be ruled out in one contact investigation due to different Pango lineages in the index and contact (ID 30). We were unable to study transmission in the remaining pairs, since WGS failed in two secondary cases due to low viral RNA content in the sample (ID 28; 41), and one index case (ID 33) did not consent to participate.

The phylogenetic analysis showed that SARS-CoV-2 viruses were more similar within than between schools, with some exceptions (see Figure 1). Analysis of the B.1.367 linage showed ≤2 SNPs difference among the index cases (ID-03; 04; 06) from three schools in neighboring communities in September and October 2020. This probably reflects indirect, rather than direct, transmission since this SARS-CoV-2 variant dominated in this geographic area during the specific time period. Similar interpretations can be assigned to the B.1.36.21, B.1.1.305 and K.3 lineages, where index cases were identified with similar viruses (≤2 SNP differences) close in time.

### 3.4. Study Data Compared with Surveillance Data

Linkage of the study data with national laboratory data revealed that 63% (434) of the participants had test results from testing outside of the study within two weeks prior to the date of the first saliva sample and four weeks after the date of the second saliva sample; 56% (333) in the pre-Alpha period versus 93% (101) in the Alpha-period. The large difference between the periods reflects the change in the recommendations and easier access to and availability of tests in the Alpha period. Concordance of test results from the two sources was high, at 98% (427/434). Among the seven individuals with discordant results, two were index cases with negative saliva tests; one had been drinking milk before the saliva collection, and the other had low viral RNA content in the original sample. Among the five contacts, four had a negative test reported from the study or MSIS 3–10 days prior to a positive result, which may reflect a natural course of the disease or a later exposure. One contact tested negative one month after a positive test result.

Data from the municipalities show that they identified five additional SARS-CoV-2 positive cases among contacts who had not consented to participate in the study; four children (primary school) and one adult (secondary school), in total 12 positive cases out of a total of 935 (1.3%, 95%CI 0.67–2.23) contacts; nine in the pre-Alpha period and three in the Alpha period. It was not possible to distinguish between primary and secondary cases among these. Comparison of infection rates in contacts not enrolled and those enrolled in the study was not significantly different, as determined by using Fisher’s exact test (*p*-value ≤ 0.05).

### 3.5. Mitigation Measures in School

Data on IPC-measures is available from 39 of the 43 schools. Overall, mitigation measures were well implemented and consistent with national guidelines (Appendix A
Table A2). The median IPC score was 7 (range 6–8). The median score was higher in the Alpha than the preAlpha-period, Table 1. More details are available in Appendix A
Table A3. Nine schools reported insufficient physical distancing of children indoors; three due to shared indoor spaces, four due to the mixing of child cohorts and two schools because they are based on an educational model in which children engage in more self-directed group work. Six schools reported insufficient physical distancing of staff, due to shared office space across staff cohorts and physical attendance in meetings. Promotion of a work-from-home policy when possible was the measure least likely to be implemented. This was only implemented in 18% (7/39) of the schools, and mainly in the second half of the academic year only (Appendix A
Table A3).

## 4. Discussion

In this large-scale prospective study with rigorous follow-up and testing of all school contacts, we found only limited child-to-child and child-to-adult SARS-CoV-2 transmission in schools, and even lower rates when WGS was used to study possible transmission links. This holds also for the more transmissible Alpha variant. The study was carried out over nine months, covering the second and third COVID-19 waves in Norway. Since we excluded contact investigations with adult cases that were known at the time of enrolment, our data provide a valid estimate on the role of children in transmission of SARS-CoV-2 in schools.

A high number of primary cases could have indicated multiple introductions of SARS-CoV-2 into the school or ongoing asymptomatic infection prior to the diagnosis of the index case. However, few primary cases were detected and all with sequencing information available had matching SARS-CoV-2 Pango lineages with the corresponding index case. Little data are available to compare the primary and secondary infection rates in this study, while the overall infection rate of 1.4% among child contacts is similar to the 0%– 3% reported by others [20,21,22,23,24,25]. A recent meta-analysis, which was restricted to studies in which all contacts were tested (available as a Lancet preprint), reported a pooled secondary attack rate from child index cases in eight schools at 0.5% (95% CI: 0.1–0.16) [23]. Data, including a subset of data published as rapid communication from the current study [7], were mostly from the pre-Alpha period. Their pooled estimate is lower than the overall infection rate reported in our study. However, the number of contacts per case differed widely between studies (range 10–297), which may have impact on secondary attack rates.

Only one out of 100 adult contacts tested positive for SARS-CoV-2. Low child-to-adult transmission in school-settings is also reported by others [4,20,24]. These findings are underpinned by population-based studies showing that educational staff are more likely to transmit the disease to students than vice-versa [26], and that the incidence among teachers is comparable to the population of the same age [27].

The Alpha variant has been shown to be more transmissible compared to earlier variants in all age groups [28]. Yet, its role in transmission in school settings is not clear. A study from England suggested that the rise in school absences observed in secondary schools in England in December 2020 in the areas where the Alpha variant first emerged, followed a preceding rise in COVID-19 incidences in the affected communities, therefore reflects higher community incidences rather than in-school transmission [29]. Several studies have pointed to the interdependency between the community infection level and infections in schools [4,30]. We found negligible differences in primary and secondary infection rates in the pre-Alpha and Alpha periods. However, the number of included contact-investigations were lower in the Alpha period; half of them were in 1–4th grade, where secondary attack rates are reported to be lower [4]. In addition, many schools were on the red IPC level with smaller cohorts and more comprehensive mitigation measures at the time that the Alpha-variant established itself in Norway. On September 3rd, 2021, the European Centre for Disease Control downgraded the Alpha variant from a variant of concern to a de-escalated variant, due to its limited impact on vaccine-induced immunity and the drastically reduced circulation following the emergence of the Delta variant (B.1.617.2) [31]. The Delta variant established itself in Norway during summer, when schools were closed, and is therefore not reflected in our study results.

Overall, WGS of the virus revealed high consistency in the variants between index cases and the corresponding primary and secondary cases, indicating that the cases were related, either through direct transmission or that they were infected from a common source. However, the direction of transmission is not necessarily clear. Being identified as the index case may simply reflect that they were the first to be tested. We also found indexes in separate contact investigations to have matching Pango lineage variants with ≤2 SNPs differences. Although direct transmission outside the school setting cannot be ruled out, this likely reflects indirect community transmission.

Home sampling of saliva is susceptible to poor sampling procedures. Parents were given face-to-face detailed instructions on how to support their children in providing saliva samples at home. Reassuringly, we found very high consistency between the saliva test results in the study and results from routine nasopharyngeal swabs in the community.

Mitigation measures were well implemented overall, although less comprehensively for staff than for children. Measures to reduce the number of contacts and to maintain distance among the remaining contacts will inevitably prevent transmission. However, evidence on the effectiveness of mitigation measures are scarce and somewhat heterogeneous [32]. Measures are commonly implemented as packages, making it difficult to disentangle the effect of the individual interventions. A recent modelling study reported a dose-dependent relationship between the number of mitigation measures implemented in schools and the COVID-19 incidence in households; when ≥7 mitigation measures were implemented there was no significant association between the two [33]. When treated as independent effects, daily symptom screening, teacher mask mandates and cancelling of extra-curricular activities were associated with greater reductions than the average [33]. The use of masks has not been recommended in the classroom setting in Norway for staff or students.

The study has contributed to the policy in Norway where primary schools have remained open with strict mitigation measures throughout the study period, and secondary schools have only had limited periods of distant learning in periods of high community transmission.

The strengths of this study include the large number of contact investigations with comprehensive data and testing of all enrolled contacts. The study covered low- and high-incidence periods, including the period before and after the Alpha variant emerged. Reassuringly we found similar infection rates in the study, as reported by the municipalities, implying non-selective participation. Further, the saliva test results showed high concordance with surveillance data, implying that our results were valid. WGS improved the validity of the results as it allowed us to verify whether the index and related contacts were infected by an identical (≤2 SNPs) strain, as well as excluding transmission in one contact investigation. This could have been misinterpreted by solely using PCR-based typing.

We did not include consecutive contact-investigations in schools following primary- or secondary cases, which may lead to underestimation of our study findings. The results are not necessarily directly applicable to new variants of SARS-CoV-2, such as the highly transmissible Delta variant (B.1.617.2 and AY.x) and new variants which may emerge. These studies are also important to conduct for new variants, to ensure that the measures to control transmission are well balanced against educational needs for children.

## 5. Conclusions

We found limited child-to-child and child-to-adult transmission of SARS-CoV-2 in schools, which also holds for the Alpha variant. Since we excluded contact investigations with adult cases known at the time of enrollment, our data provide a valid estimate on the role of children in transmission of SARS-CoV-2 in schools. Looking only at outbreaks in schools in general, without considering contribution of adults in transmission, may overestimate the role of children in transmission.

## Figures and Tables

**Figure 1 microorganisms-09-02587-f001:**
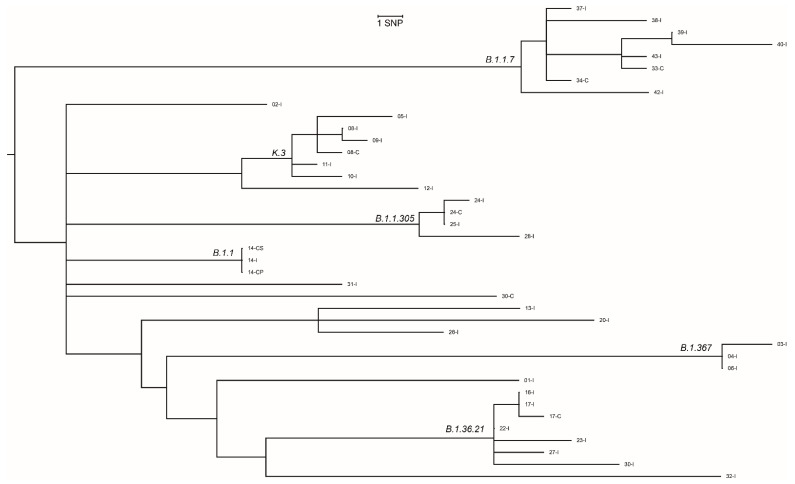
Maximum likelihood phylogenetic tree of SARS-CoV-2 strains from cases included in contact investigations (n, 33), Oslo and Viken, August 2020–May 2021; SNP; single nucleotide polymorphism; the whole genome sequencing failed due to low viral load in two of the secondary cases, and one the index case did not consent to participate.

**Table 1 microorganisms-09-02587-t001:** Characteristics of the study population in 43 contact investigations in Oslo and Viken, August 2020 to May 2021.

	Whole Study Period	Pre-Alpha	Alpha
Number of contact investigations, all (*n*, %)	43	32	11
1–4th grade	15 (35)	9 (28)	6 (55)
5–7th grade	14 (33)	11 (34)	3 (27)
8–10th grade	14 (33)	12 (38)	2 (18)
Number of child contacts, all (*n*, %)	599	441	118
1–4th grade	214 (36)	154 (35)	60 (51)
5–7th grade	188 (31)	166 (38)	22 (19)
8–10th grade	157 (26)	121 (27)	36 (31)
Number of adult contacts, all (*n*, %)	100	81	19
1–4th grade	53 (53)	43 (53)	10 (53)
5–7th grade	20 (20)	17 (21)	3 (16)
8–10th grade	27 (27)	21 (26)	6 (32)
Number of child contacts per index case (median, range) ^#^	13 (3–33)	15 (4–33)	6 (4–32)
Participation child contacts (median, range) ^#^	60 (14–89)	68 (14–89)	44 (14–72)
Number of adult contacts per index case (median, range) ^#^	2 (0–12)	2 (0–12)	1 (0–3)
Participation adult contacts (median, range) ^#^	67 (0–100)	67 (0–100)	50 (0–100)
IPC-score (median, range)	7 (6–8)	6 (5–8)	7 (6–8)

IPC-score, Infection Prevention and Control (mitigation) measures, scored on a scale from 0–8 with higher scores indicating higher consistency with national guidelines; pre-Alpha and Alpha reflects the period before and after the Alpha variant (B.1.1.7) was established (August–February versus March–May); *^#^* Participation was calculated as the percentage of included contacts of all identified contacts per contact-investigation, presented as median (%) and range (%–%).

**Table 2 microorganisms-09-02587-t002:** Primary, secondary and all infected child contacts detected in school contact investigations (n, 43) within 7–10 days following the last exposure, Oslo and Viken, August 2020–May 2021.

	Number of Contacts, *n* (*n*) ^#^	Primary ^I^ Cases, *n* (%, 95% CI)	Secondary ^II^ Cases, *n* (%, 95% CI)	All Infected Cases ^III^ *n* (%, 95% CI)
All child contacts	559 (537)	4 (0.7, 0.20–1.82)	4 (0.7, 0.20–1.90)	8 (1.4, 0.62–2.80)
pre-Alpha *	441 (426)	3 (0.7, 0.14–1.97)	3 (0.7, 0.15–2.04)	6 (1.4, 0.50–2.94)
Alpha *	118 (111)	1 (0.9, 0.02–4.63)	1 (0.9, 0.02–4.92)	2 (1.7, 0.21–5.99)
Child contacts by school grade				
Child contacts, grade 1–4 (all)	214 (206)	1 (0.5, 0.01–2.58)	0	1 (0.5, 0.01–2.58)
pre-Alpha	154 (148)	1 (0.7, 0.02–3.56)	0	1 (0.7, 0.02–3.56)
Alpha	60 (58)	0	0	0
Child contacts, grade 5–7 (all)	188 (183)	1 (0.5, 0.01–2.93)	2 (1.1, 0.13–3.89)	3 (1.6, 0.33–4.59)
pre-Alpha	166 (163)	1 (0.6, 0.02–3.31)	1 (0.6, 0.02–3.37)	2 (1.2, 0.15–4.28)
Alpha	22 (20)	0	1 (5.0, 0.13–22.87)	1 (4.5, 0.12–22.84)
Child contacts, grade 8–10 (all)	157 (148)	2 (1.3, 0.15–4.53)	2 (1.4, 0.16–4.58)	4 (2.6, 0.70–6.39)
pre-Alpha	121 (115)	1 (0.8, 0.02–4.52)	2 (1.8, 0.20–6.14)	3 (2.5 0.51–7.07)
Alpha	36 (33)	1 (2.8, 0.07–14.5)	0	1 (2.8, 0.07–14.53)

* pre-Alpha and Alpha reflects the period before and after the Alpha variant (B.1.1.7) was established (August–February versus March–May); ^#^
*n* (*n*) number of contacts, the parentheses represents the number of contacts with a second saliva sample; 18 (3%) child contacts did not provide a second saliva sample and primary cases were excluded; ^I^ a contact of the index case who tested positive for SARS-CoV-2 in the first saliva sample; ^II^ a contact of the index case who tested positive for SARS-CoV-2 in the second saliva sample, following a first negative test; ^III^ a contact who tested positive for SARS-CoV-2 in either of the two saliva tests, thus including both the primary and secondary cases.

**Table 3 microorganisms-09-02587-t003:** SARS-CoV-2 Pango lineage variants among child index cases (n, 33), Oslo and Viken, August 2020–May 2021.

SARS-CoV-2 Variant ^I^	Number of Contact Investigations	Time of Contact-Investigation	Period ^#^
B.1.1.7	7	April 2021–May 2021	Alpha
B.1.36.21	6	November 2020–January 2021	pre-Alpha
K.3	5	October 2020	pre-Alpha
B.1.367	3	September 2020–October 2020	pre-Alpha
B.1.1.305	3	December 2020–January 2021	pre-Alpha
B.1.1	2	November 2020–January 2021	pre-Alpha
Other *	7	September 2020–January 2021	pre-Alpha

For one index case, the whole genome sequencing failed due to low viral RNA content in the sample; ^I^ Pango lineage nomenclature https://cov-lineages.org/index.html, accessed on 28 September 2021; ^#^ Period; pre-Alpha and Alpha reflects the period before and after the Alpha variant (B.1.1.7) was established (August–February versus March–May); * The following SARS-CoV-2 lineages occurred in only one index case each: B.1, B.1.1.39, B.1.1.277, B.1.177.44, B.1.177.82, B.1.177, and B.1.36.1.

**Table 4 microorganisms-09-02587-t004:** SARS-CoV-2 variants in contact investigations in which primary and secondary cases were detected (n, 8), Oslo and Viken, August 2020–May 2021.

ID	Period	School-Grade	SARS-CoV-2 Variant			SNP Difference
			Index Case ^I^	Primary Case ^II^	Secondary Case ^III^	
08	pre-Alpha	1–4	K.3	K.3		2 SNPs
14	pre-Alpha	5–7	B.1.1	B.1.1	B.1.1	0 SNP
17	pre-Alpha	8–10	B.1.36.21	B.1.36.21		1 SNP
24	pre-Alpha	5–7	B.1.1.305	B.1.1.305		1 SNP
28	pre-Alpha	8–10	B.1.1.305		WGS failed	na
30	pre-Alpha	8–10	B.1.36.21		B.1.1.333	na
33	Alpha	8–10		B.1.1.7		na
41	Alpha	5–7	B.1.1.7		WGS failed	na

ID; identifier for each contact-investigation (reference, Appendix A
Table A2); Period; pre-Alpha and Alpha reflects the periods before and after the Alpha variant (B.1.1.7) was established (August–February versus March–May); SNP; Single nucleotide polymorphism; ^I^ The index case was the child first identified with confirmed SARS-CoV-2 leading to the contact-investigation; ^II^ A contact of the index case who tested positive for SARS-CoV-2 in the first saliva sample; ^III^ A contact of the index case who tested positive for SARS-CoV-2 in the second saliva sample, following a first negative test.

## Data Availability

The data presented in this study are available on request from the corresponding author. De-identified data are not publicly available due to regulations in the Norwegian Health Research Act and the Norwegian Data Protection Act for use (and storage) of Personal Data related to health. To receive access to the data the applicant will need to provide an ethical approval from their IRB or equivalent body, and from Regional Committees for Medical and Health Research Ethics in Norway.

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
