# Peer review of "Comprehensive Contact Tracing, Testing and Sequencing Show Limited Transmission of SARS-CoV-2 between Children in Schools in Norway, August 2020 to May 2021"

_microorganisms, 2021, doi:10.3390/microorganisms9122587_

Round 1

Reviewer 1 Report

Authors addressed a still relevant issue of the role of school children in transmission of SARS-CoV-2. The important advantage of this study is the use of GWS analysis to confirm possible within-school transmission. The manuscript is well written and easy to follow despite complex analysis performed.

There are few suggestions that could improve the quality of the paper before its acceptance:

Dividing contacts in primary and secondary would have much sense if contact tracing was the subject of investigation, this way it is used only in the descriptive table and disappears afterwards.

Testing of all contacts is mentioned as one of the strengths of the study while in results 60% of children contacts and 63% of adult contacts accepted to participate in the study, which, being almost half of contacts, is closer to selection bias than to a strength of the study.

4 primary and secondary cases for which transmission was confirmed by means of GWS analysis belonged to which type of school?

Layout of table 2 should be improved to facilitate faster understanding that rows with “Child contacts, grade…” represent subgroups of previous rows as they sum up to 100%.

Author Response

We thank the reviewer for the positive feedback and the constructive comments. Below, a point-by-point response is given to the concerns raised by the reviewer.

  1. Dividing contacts in primary and secondary would have much sense if contact tracing was the subject of investigation, this way it is used only in the descriptive table and disappears afterwards.

Response: We thank the reviewer for this comment. We have added a paragraph on this in the Discussion, lines 304-308

  1. Testing of all contacts is mentioned as one of the strengths of the study while in results 60% of children contacts and 63% of adult contacts accepted to participate in the study, which, being almost half of contacts, is closer to selection bias than to a strength of the study.

Response: We agree that the study did not enrol all identified school contacts and that this may have introduced bias if the decision to participate was selective. However, throughout the study we systematically obtained aggregated data from the municipalities on the outcome of their contact-investigations in which all contacts were included. Although some cases were missed in our study due to non-participation, the overall infection rates were similar in the study and in the municipalities. This, alongside the prospective design makes selection-bias less likely, which is also in line with a prospective study design. This is already described in Methods, lines 151-155, in the Results, lines 282-283, and in the Discussion, lines 374-375. We added more detail in Result, lines 282-283, and we amended the wording in the Strengths of the study to include “all enrolled contacts, line 373

  1. 4 primary and secondary cases for which transmission was confirmed by means of GWS analysis belonged to which type of school?

Response: Among the four primary cases for which transmission was confirmed, one belonged to 1-4th grade, two to 5-7th grade and one to 8-10th grade.  This information is already presented In Table 4 (page 7). We also refer to the table in the txt.

  1. Layout of table 2 should be improved to facilitate faster understanding that rows with “Child contacts, grade…” represent subgroups of previous rows as they sum up to 100%

Response: We have added a line in Table 2 (page 5 and 6) including a heading stating “Child contacts by school grade” and we hope this will facilitate faster understanding.

Reviewer 2 Report

Winje et al. performed an extensive contact tracing and testing of SARS-CoV2 between children and contact persons in Norway. Overall the study seems to be performed at a high quality and the results are carefully interpreted. The manuscripts seems ready for publication.

Author Response

Thank you for your positive comments to our manuscript. 

Reviewer 3 Report

The aim of the study was to investigate a very current and interesting issue of SARS-CoV-2 spread using e.g. contact tracing and WGS technique. The study seems to be well planned, organized and conducted. I suppose a lot of effort went into the study. Several findings made on this basis, in my opinion, are really relevant and interesting for the readers.

My only concerns are:

It should be specified in “Materials and methods” section, which particular assay/assays was/were applied for the samples testing and what were the exact results interpretation criteria of the kit.

In my opinion, the failed WGS should not be explained by low viral RNA concentration (Ct>31 should be still sufficient for the study purpose). Did you re-test the original sample? Maybe it was a cross-contamination only at the initial testing step?

The title might be re-written to sound like a more general observation, in my opinion.

In addition, at least in my opinion, some of the sentences from the "Discussion" section should be moved to the Introduction part, e.g. lines 327-328.

Moreover, there are some minor linguistic, grammatical and punctuation errors in the manuscript that have to be corrected before the publication.

However, in general, the above-mentioned issues do not reduce the substantive value of the study.

Author Response

We thank the reviewer for the positive feedback and the constructive comments. Below, a point-by-point response is given to the concerns raised by the reviewer.

  1. It should be specified in “Materials and methods” section, which particular assay/assays was/were applied for the samples testing and what were the exact results interpretation criteria of the kit.

Response: As suggested by the reviewer, we have added information on the particular assay/assays that was used, lines 113-125

  1. In my opinion, the failed WGS should not be explained by low viral RNA concentration (Ct>31 should be still sufficient for the study purpose). Did you re-test the original sample? Maybe it was a cross-contamination only at the initial testing step?

Response: We checked the sample again and noticed that the Ct value of the target genes were > 33 and not > 31. This is now corrected in the manuscript (line 221). The SARS-CoV-2 positive sample from this specific index case was whole genome sequenced once, however the sequencing was not successful and we only achieved a few thousand reads. This is in concordance with sequencing of other SARS-CoV-2 positive samples with Ct values >33 at our institute, in which the majority of the samples fail to have a sequence.

  1. The title might be re-written to sound like a more general observation, in my opinion.

Response: We acknowledge the reviewers view and appreciate the feedback, However, we suggest to keep the title as is, since this school-transmission study differs from others in the comprehensive testing, data-collection and the sequencing to enlighten the results. We hope this is acceptable for editors and the reviewer.

  1. In addition, at least in my opinion, some of the sentences from the "Discussion" section should be moved to the Introduction part, e.g. lines 327-328.

Response: We have moved the above-mentioned sentence from the Discussion to the Introduction, line 65-66.

  1. Moreover, there are some minor linguistic, grammatical and punctuation errors in the manuscript that have to be corrected before the publication.

Response: We have read the manuscript carefully and corrected linguistic, grammatical and punctuation errors. We hope this is now correct. See track-changes.